# Intermarriage and COVID-19 mortality among immigrants. A population-based cohort study from Sweden

Siddartha Aradhya ,[1] Maria Brandén,[1,2] Sven Drefahl,[1] Ognjen Obućina,[3] Gunnar Andersson,[1] Mikael Rostila,[4] Eleonora Mussino,[1] Sol Pía Juárez [4]

[1]Stockholm University Demography Unit (SUDA), Department of Sociology, Stockholm University, Stockholm, Sweden
[2]The Institute for Analytical Sociology (IAS), Linköping University, Linkoping, Sweden
[3]Institut national d'études démographiques (INED), Aubervilliers, France
[4]Department of Public Health Sciences, Stockholm University, Stockholm, Sweden

**Correspondence to**
Dr Sol Pía Juárez;
sol.juarez@su.se

## ABSTRACT

**Objectives** To evaluate the role of language proficiency and institutional awareness in explaining excess COVID-19 mortality among immigrants.

**Design** Cohort study with follow-up between 12 March 2020 and 23 February 2021.

**Setting** Swedish register-based study on all residents in Sweden.

**Participants** 3 963 356 Swedish residents in co-residential unions who were 30 years of age or older and alive on 12 March 2020 and living in Sweden in December 2019.

**Outcome measures** Cox regression models were conducted to assess the association between different constellations of immigrant-native couples (proxy for language proficiency and institutional awareness) and COVID-19 mortality and all other causes of deaths (2019 and 2020). Models were adjusted for relevant confounders.

**Results** Compared with Swedish-Swedish couples (1.18 deaths per thousand person-years), both immigrants partnered with another immigrant and a native showed excess mortality for COVID-19 (HR 1.43; 95% CI 1.29 to 1.58 and HR 1.24; 95% CI 1.10 to 1.40, respectively), which translates to 1.37 and 1.28 deaths per thousand person-years. Moreover, similar results are found for natives partnered with an immigrant (HR 1.15; 95% CI 1.02 to 1.29), which translates to 1.29 deaths per thousand person-years. Further analysis shows that immigrants from both high-income and low-income and middle-income countries (LMIC) experience excess mortality also when partnered with a Swede. However, having a Swedish-born partner is only partially protective against COVID-19 mortality among immigrants from LMIC origins.

**Conclusions** Language barriers and/or poor institutional awareness are not major drivers for the excess mortality from COVID-19 among immigrants. Rather, our study provides suggestive evidence that excess mortality among immigrants is explained by differential exposure to the virus.

International evidence has shown that immigrants and ethnic minorities are disproportionately at risk of severe COVID-19 complications and death.[1–13] In the context of an ongoing pandemic, an effort to understand the causes for why some groups are more affected is a public health priority.[14–17]

### Strengths and limitations of this study

► This study uses total population data with all deaths (from COVID-19 and other causes) in Sweden from 12 March 2020 to 23 February 2021.
► We identified the origin of co-resident couples to evaluate the role of language proficiency and institutional awareness (eg, healthcare system) in explaining excess COVID-19 mortality among immigrants.
► We compare COVID-19 mortality with all other causes of death during the pandemic and all-cause mortality 1 year prior to evaluate the relative impact of the pandemic within each group.
► The analyses do not include information on occupation, however the most vulnerable individuals are beyond retirement age.

Among these groups, excess mortality has been suggested to be the result of differential exposure (eg, high-risk occupations or overcrowded accommodation), susceptibility (eg, pre-existing conditions) and language barriers and access to healthcare.[18–20] Recent studies, however, suggest that immigrants and minorities maintain an excess mortality even after controlling for socioeconomic status and housing conditions.[3 21] Yet there is paucity of evidence on the role of language barriers and institutional awareness in explaining the COVID-19 excess mortality experienced by immigrants.

Sweden took a distinct approach to dealing with the COVID-19 pandemic as compared with other Western countries by not implementing lock downs or mask mandates and instead relied largely on recommendations. The authorities justified implementing a relatively less rigid approach by arguing that Swedes have a high level of trust in their institutions and as such follow governmental recommendations.[22 23] The strategy relied primarily on public health advice (regarding hygiene routines, social distancing and suspension from work, school or daycare in case of minor

symptoms) *in lieu* of mandates which are not permitted under Swedish law.[22] The effectiveness of the adopted strategy strongly depends on the ability of all members of the society to understand the recommendations, which is a basic condition for their adherence. Under this rationale, it is unsurprising that excess mortality observed among immigrants[24]—especially concentrated among those with more distant origins—could be interpreted as a consequence of lower adherence to recommendations and/or related factors.

More specifically, it has been argued that immigrants may, as a result of inadequate language proficiency and institutional awareness, have a poor understanding of the healthcare system and of the recommendations during the COVID-19 pandemic putting them at higher risk of exposure and vulnerability to the virus. Analysing intermarried immigrants present a unique opportunity to generate evidence for this explanation.

Within the field of immigrant integration, intermarriage has been considered the ultimate stage of acculturation[25–27] and is, both, a marker of and facilitator for integration. Marrying a native is strongly related to language abilities, knowledge of the host country's institutions and social practices as well as the ethnic composition of one's social circle. As a result, intermarriage reflects the narrowing of sociocultural distance between ethnic Swedes and immigrants, which renders it an ideal measure to evaluate the role of understanding and awareness of recommendations from Swedish authorities as explanations for the excess COVID-19 mortality among immigrants.

If understanding and awareness of recommendations (language *in primis*) explain the excess mortality, immigrants partnered with a Swede and, in particular, Swedes partnered with an immigrant should have similar mortality to Swedes partnered with a Swede. To this end, the aim of this study is to examine the association between sociocultural integration and COVID-19 mortality by examining native-immigrant couple dyads—a well-regarded measure that has been shown to be a marker and facilitator of integration.

## METHODS
### Study population
An observational cohort study was conducted using Swedish register data. The study includes all Swedish residents who were 30 years of age or older and were cohabiting with another adult who was at least 30 years of age and alive on 12 March 2020, residing in Sweden in December 2019 (n=4 019 418). This age restriction was established to ensure co-resident individuals were family members and not flat mates. The follow-up period was 12 March 2020 up until 23 February 2021. We excluded individuals who had not lived in Sweden in the two prior years (n=40 515), because they could not be linked to all records of data. In addition, we excluded individuals with missing data on country of birth (n=142) and income (n=15 405) of either partner. The final study population consists of 3 963 356 individuals (18.5% immigrants) (figure 1).

### Patient and public involvement
No patient was involved.

### Data
We use information from several Swedish administrative registers linked through personal identity numbers that are unique to each person with legal residence in Sweden. Data on deaths were retrieved from the Cause of Death Register. Socioeconomic and demographic variables (income, education, number of children and region of residence) were drawn from the Longitudinal Integrated Database for Health Insurance and Labour Market Studies (LISA), and residential information (type and crowdedness of the dwelling) were drawn from the

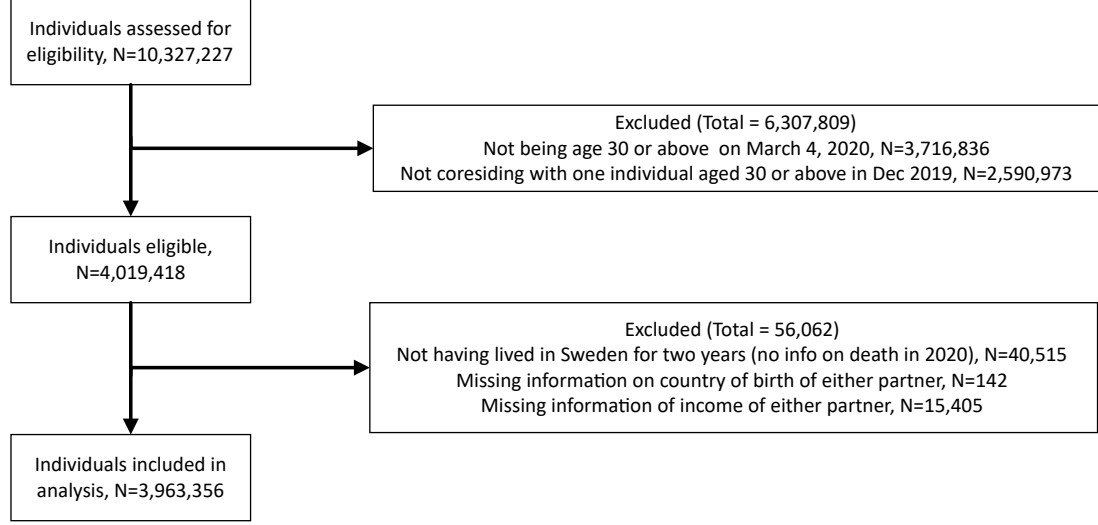

**Figure 1** Selection flow and final sample.

Dwelling Register. All covariates in our study are time-constant and either measured at the end of 2019 (all variables at the household and neighbourhood level) or 2018 (highest education attained, sum of the individual net incomes of the two co-resident adults, total number of individuals in the household under 30). Information on age, sex, country of birth and immigrant density in the neighbourhood stem from the Total Population Register. It is important to note that all individuals registered in Sweden are entitled to healthcare access.

## Study variables

COVID-19 mortality was identified by the Swedish National Board of Health and Welfare (*Socialstyrelsen*), the agency responsible for the Cause of Death Register. COVID-19 mortality was identified using the following International Classification of Diseases (ICD) codes for the underlying cause of death: U07.1 (3915 deaths), U07.2 (127 deaths) and B34.2 (2 deaths); for 522 more deaths ICD codes U07.1, U07.2 or B34.2 were listed as contributing causes of death, excluding mortality from all other causes of death (30 374 deaths). Given the timeliness of the data, the assignment of the underlying cause of death should be understood as preliminary.

Immigrant-native couple types were created by combining information from the Dwelling and the Total Population Registers to create the couple type which include two individuals of at least 30 years of age co-residing in the same household. The variable is classified into the following four ego-partner categories using information on country of birth: (i) native-native, (ii) native-immigrant, (iii) immigrant-native and (iv) immigrant-immigrant. We chose (i) native-native couples as the reference group for our analyses as they (a) constitute the largest category among all groups considered and (b) represent the institutional awareness and language of the host population from which we expect other groups to deviate. We further disaggregated the groups by immigrant's origins defined according to the World Bank classification based on the Gross National Incomes per capita using the WB Atlas method[28] as high-income countries (HIC) and low-income and middle-income countries (LMIC).

We derived individual income and calculated the sum of the two partners' net incomes (household), categorised into tertiles based on all adult residents of Sweden. We derived education data from Swedish Educational Registers and categorise our population into four categories: those with primary schooling, secondary schooling, post-secondary education and those with missing information on education. Missing information on education is generally very low but 88% of those with missing education are immigrants. We additionally performed multiple imputation to test how the missing values for education impact our results (see online supplemental appendix figure S1). In addition, we dropped all missing categories to assess whether the distributions of covariates were affected by missing values (see online supplemental appendix table S1). From the Swedish Dwelling Register, we accessed information on size of the dwelling and a unique dwelling code which enables us to link individuals who live together in a household and determine co-residence. In addition to define couples, we use this information to create: the number of individuals per square metre in the household (with a separate category for a small group of individuals, due to missing information on square metres in some detached houses), the number of individuals living in the household under the age of 30 and dwelling type (multi-family, single family or care home). We include in our model also the share of immigrants in the local neighbourhood, DeSO (a smaller subdivision of administrative areas based on demographic characteristics produced by the Swedish administrative statistics).

## Statistical analysis

We conducted Cox proportional hazards regressions (using age as the timescale) to estimate HR and 95% CI for the association between immigrant-native couple type and COVID-19 mortality. Individuals exited the study by (1) dying between 12 March 2020 and 23 February 2021, or (2) being alive on 23 February 2021. We estimated two separate regressions estimating the cause-specific hazard of dying from COVID-19, right-censoring all individuals that die from other causes and (2) the cause-specific hazard of dying from other causes than COVID-19, right-censoring all individuals that die from COVID-19. In addition, we conducted Cox regressions for dying from all causes of death that occurred between 12 March 2019 and 23 February 2020 (31 653 deaths)—the same period we observe COVID-19 deaths 12 contiguous months prior to the start of the pandemic. Since mortality from COVID-19 and other causes of deaths during the study window are not fully independent of each other, our estimates for all-cause mortality between 12 March 2019 and 23 February 2020 were used to evaluate the robustness of our estimates for mortality during the pandemic. In addition, the comparison between all-cause mortality 1 year prior and mortality from causes other than COVID-19 during the pandemic, by immigrant-native couple type, will allow us to examine whether the latter has a distinctive role in relation to the pandemic.

Two models were estimated: (1) a simple model with age as the time scale adjusted for sex and (2) the same model with further adjustments. In the latter analysis, we adjust for education, neighbourhood immigrant density and region of residence fixed-effects that are confounders, as well as factors that are on the causal pathway that have been previously used to explain the excess mortality of immigrants (ie, number of individuals in the household below the age of 30 years, dwelling type, square metres per person in the dwelling, household income). In addition, we conducted two sensitivity analyses. One in which we examined the partner's origin among Swedish-migrant partnerships (HIC or LMIC) to check whether patterns are consistent across groups and, a second, in which we excluded all individuals born in Sweden with at least one

foreign-born parent. All analyses were conducted using Stata Statistical Software: Release 16 (StataCorp, College Station, Texas, USA).

## RESULTS

During the 3 759 610 person-years of observation, 4564 COVID-19 deaths occurred in our study population between 12 March 2020 and 23 February 2021. Table 1 shows the distribution of population at risk and deaths by all covariates. In our population, 6.09% of individuals are in immigrant-native couples. Native-native couples show the lowest deaths per thousand person-years (1.18/1000), whereas all other couple types show higher death rates of COVID-19 mortality (approximately 1.3–1.4 per thousand person-years). Of the study population, 6.55% of COVID-19 deaths were attributed to native-immigrant mixed couples and 14.04% to immigrant-immigrant couples.

Figure 2 displays mortality risks from COVID-19, all other causes of death during the pandemic and all-causes of death 1 year prior across couple types with native-native as the reference (see online supplemental appendix table S2 for estimates). Panel A presents models adjusted for age and sex and panel B presents the estimates including adjustments. In A, individuals in immigrant-immigrant couples show the highest HR of dying from COVID-19 (HR 2.47; 95% CI 2.27 to 2.69) and those in native-native couples the lowest (reference group) while natives (HR 1.40; 95% CI 1.25 to 1.58) and immigrants (HR 1.50; 95% CI 1.33 to 1.69) in mixed couples showed intermediate mortality levels. All causes of death in the year prior to the pandemic and all other causes of death during the pandemic show little differences between couple constellations. After adjustments (panel B), differences across groups attenuate, but the gradient in COVID-19 mortality remains. Individuals in immigrant-immigrant couples still display the highest HR of dying from COVID-19 relative to the reference group (HR 1.43; 95% CI 1.29 to 1.58), followed by those in immigrant-native couples (HR 1.24; 95% CI 1.10 to 1.40) and native-immigrant couples (HR 1.15; 95% CI 1.02 to 1.29). It is important to note that there is no statistically significant difference in the risk of COVID-19 mortality between immigrant-native and immigrant-immigrant couples. An opposite gradient is observed with respect to all-cause mortality in the year prior to the pandemic and all other causes of death during the pandemic, with immigrant-immigrant couples displaying the lowest mortality.

Figure 3 is an extension of figure 2 disaggregating the immigrant population by income level of their country of birth (see online supplemental appendix table S3 for estimates). In panel A, there are elevated HRs across all origin groups relative to native-native couples. Immigrants in LMIC-immigrant couples display the highest HR relative to the reference group (HR 3.60; 95% CI 3.25 to 3.99) followed by those in LMIC-native couples (HR 1.91; 95% CI 1.42 to 2.57). Natives in native-immigrant

couples, where HIC and LMIC have been pooled (HR 1.41; 95% CI 1.25 to 1.58), HIC-native (HR 1.44; 95% CI 1.27 to 1.64) and immigrants in HIC-immigrant (HR 1.54; 95% CI 1.34 to 1.77) couples display relatively similar HRs. After adjustment (panel B), all groups with an immigrant still display higher HRs relative to native-native couples, but at a lower level. LMIC-immigrant couples experience a particularly strong reduction in their HRs (HR 1.84; 95% CI 1.62 to 2.09). Compared with LMIC-native couples, their HR remains slightly higher (HR 1.35; 95% CI 1.00 to 1.82). We also find that HIC-native (HR 1.23; 95% CI 1.08 to 1.40) and HIC-immigrant (HR 1.11; 95% CI 0.97 to 1.28) couples display higher HRs relative to the reference group.

Sensitivity analyses showed that disaggregating the origin of immigrants in native-immigrant partnerships (online supplemental appendix figure S2) and excluding second-generation Swedes from the Swedish-born population (online supplemental appendix figure S3) produce no further differentials in mortality risks.

## DISCUSSION

Our study shows that immigrants have excess COVID-19 mortality regardless of the origin of their partner, where having a Swedish-born partner is only partially protective against COVID-19 mortality among immigrants from LMIC. These findings challenge hypotheses that poor language proficiency and institutional awareness are major contributing factors explaining the excess mortality from COVID-19 among immigrants.

Language has been considered a vital component of integration and relevant for accessing information for other types of medical treatments and health outcomes.[29] Given that the COVID-19 pandemic is a unique occurrence that was accompanied by a global diffusion of information, one can argue that even those with presumably little Swedish proficiency have been exposed to recommendations offered in their native languages from either public health officials from their countries of origin or via other international channels. In fact, the information that they may have received from international sources may be more relevant for specific immigrant populations, for example, how to best protect oneself when observing cultural or religious practice. At the same time, the Swedish authorities have translated information to other languages, although not culturally adapted. However, in the first stage of the pandemic, the authorities were delayed in releasing information in all languages and, at least in Stockholm, we found no group differences among intermarried groups and immigrant-immigrant couples (see online supplemental appendix figure S4). This highlights how language is no straightforward factor in mitigating the burden of COVID-19 on immigrant populations.

Prior studies in clinical settings have shown that access to a medical interpreter is associated with better health access and outcomes.[29] Although we cannot test this

**Table 1** Description of the study population, number, proportion of deaths and death rates

| | Total | | COVID-19 | | Other cause of death | | | |
|---|---|---|---|---|---|---|---|---|
| | N at 12 March 2020 | % | Deaths 12 March 2020–23 February 2021 | % | Deaths 12 March 2020–23 February 2021 | % | Exposure time in years | COVID-19 deaths per 1000 years |
| **Couple type** | | | | | | | | |
| Native-native | 2 984 648 | 75.31 | 3330 | 72.96 | 24 846 | 81.80 | 2 830 382 | 1.18 |
| Native-immigrant | 245 124 | 6.18 | 299 | 6.55 | 1739 | 5.73 | 232 563 | 1.29 |
| Immigrant-native | 241 185 | 6.09 | 294 | 6.44 | 1507 | 4.96 | 228 949 | 1.28 |
| Immigrant-immigrant | 492 399 | 12.42 | 641 | 14.04 | 2282 | 7.51 | 467 716 | 1.37 |
| **Couple type, detailed** | | | | | | | | |
| Native-native | 2 984 648 | 75.31 | 3330 | 72.96 | 24 846 | 81.80 | 2 830 382 | 1.18 |
| Native-immigrant | 245 124 | 6.18 | 299 | 6.55 | 1739 | 5.73 | 232 563 | 1.29 |
| HIC-native | 140 281 | 3.54 | 249 | 5.46 | 1294 | 4.26 | 132 929 | 1.87 |
| HIC-immigrant | 121 148 | 3.06 | 215 | 4.71 | 1170 | 3.85 | 114 765 | 1.87 |
| LMIC-native | 100 904 | 2.55 | 45 | 0.99 | 213 | 0.70 | 96 020 | 0.47 |
| LMIC-immigrant | 371 251 | 9.37 | 426 | 9.33 | 1112 | 3.66 | 352 951 | 1.21 |
| **Sex** | | | | | | | | |
| Man | 1 992 097 | 50.26 | 3145 | 68.91 | 19 003 | 62.56 | 1 887 523 | 1.67 |
| Woman | 1 971 259 | 49.74 | 1419 | 31.09 | 11 371 | 37.44 | 1 872 087 | 0.76 |
| **Education** | | | | | | | | |
| Primary | 589 969 | 14.89 | 1810 | 39.66 | 11 495 | 37.84 | 555 860 | 3.26 |
| Secondary | 1 687 832 | 42.59 | 1732 | 37.95 | 12 069 | 39.73 | 1 601 553 | 1.08 |
| Postsecondary | 1 651 905 | 41.68 | 900 | 19.72 | 6420 | 21.14 | 1 570 393 | 0.57 |
| Missing | 33 650 | 0.85 | 122 | 2.67 | 390 | 1.28 | 31 804 | 3.84 |
| **Household income (tertile)** | | | | | | | | |
| Lowest | 1 289 265 | 32.53 | 3544 | 77.65 | 22 358 | 73.61 | 1 216 124 | 2.91 |
| Middle | 1 336 367 | 33.72 | 625 | 13.69 | 5008 | 16.49 | 1 270 568 | 0.49 |
| Highest | 1 337 724 | 33.75 | 395 | 8.65 | 3008 | 9.90 | 1 272 918 | 0.31 |
| **Housing type** | | | | | | | | |
| Multifamily | 1 283 364 | 32.38 | 2248 | 49.26 | 11 797 | 38.84 | 1 216 054 | 1.85 |
| Single-family | 2 662 463 | 67.18 | 1930 | 42.29 | 16 946 | 55.79 | 2 527 860 | 0.76 |
| Care home | 17 529 | 0.44 | 386 | 8.46 | 1631 | 5.37 | 15 696 | 24.59 |
| **Number of people under 30 in the household** | | | | | | | | |
| 0 | 2 079 106 | 52.46 | 4244 | 92.99 | 28 211 | 92.88 | 1 965 558 | 2.16 |
| 1 | 614 976 | 15.52 | 180 | 3.94 | 1191 | 3.92 | 585 276 | 0.31 |
| 2 | 870 872 | 21.97 | 85 | 1.86 | 647 | 2.13 | 829 379 | 0.10 |
| 3+ | 398 402 | 10.05 | 55 | 1.21 | 325 | 1.07 | 379 397 | 0.14 |
| **m²/person in the household (crowdedness)** | | | | | | | | |
| 0- | 325 441 | 8.21 | 325 | 7.12 | 1463 | 4.82 | 309 194 | 1.05 |
| 20- | 814 282 | 20.55 | 464 | 10.17 | 2809 | 9.25 | 774 226 | 0.60 |
| 30- | 913 084 | 23.04 | 1240 | 27.17 | 7232 | 23.81 | 865 941 | 1.43 |
| 40- | 1 127 740 | 28.45 | 1662 | 36.42 | 11 264 | 37.08 | 1 068 381 | 1.56 |
| 60- | 750 691 | 18.94 | 861 | 18.87 | 7468 | 24.59 | 711 339 | 1.21 |
| Missing | 32 118 | 0.81 | 12 | 0.26 | 138 | 0.45 | 30 530 | 0.39 |
| **Share immigrants in DeSO (%)** | | | | | | | | |
| 0- | 1 306 913 | 32.97 | 958 | 20.99 | 9358 | 31.40 | 1 240 219 | 0.77 |
| 0.10- | 943 691 | 23.81 | 972 | 21.30 | 7318 | 24.09 | 895 256 | 1.09 |
| 0.15- | 659 303 | 16.63 | 858 | 18.80 | 5003 | 16.47 | 625 400 | 1.37 |
| 0.20- | 556 222 | 14.03 | 765 | 16.76 | 4575 | 15.06 | 527 391 | 1.45 |

Continued

| | Total | | COVID-19 | | Other cause of death | | Exposure time in years | COVID-19 deaths per 1000 years |
|---|---|---|---|---|---|---|---|---|
| | N at 12 March 2020 | % | Deaths 12 March 2020–23 February 2021 | % | Deaths 12 March 2020–23 February 2021 | % | | |
| 0.30- | 349 234 | 8.81 | 637 | 13.96 | 2896 | 9.53 | 331 065 | 1.92 |
| 0.50- | 147 993 | 3.73 | 374 | 8.19 | 1044 | 3.44 | 140 278 | 2.67 |
| Total | 3 963 356 | 100.0 | 4564 | 100.0 | 30 374 | 100.0 | 3 759 610 | 1.21 |

HIC, high-income country; LMIC, low-income and middle-income country.

aspect directly, our study shows that immigrants partnered with Swedes are slightly protected with respect to COVID-19 mortality. This suggests that lower language barriers may indeed be relevant with respect to interacting with the healthcare system. To the best of our knowledge, however, we are unaware of any additional provisions provided to non-Swedish speakers at hospitals or clinics during the pandemic.

In addition to disentangling the role of language barriers and lack of understanding of the healthcare system and recommendations in explaining the excess COVID-19 mortality among immigrants, our study provides suggestive evidence that the main explanation is differential exposure to the virus and not susceptibility.[20] Although it is true that Swedes partnered with a Swede show the lowest mortality, those partnered with an immigrant experience higher COVID-19 mortality. Given that Swedes do not experience language barriers or lack institutional awareness and that biological susceptibility cannot be transmitted between partners, Swedes partnered with immigrants are likely to be at either higher exposure to the virus or impacted by the social susceptibility of their partner. Swedes in mixed partnerships may be exposed to similar social environments and/or risk factors as immigrants thus placing them at a higher risk of exposure as compared to Swedes partnered with another Swede. For example, they may have more transnational contacts or are impacted by some of the same social risk factors as immigrants. Moreover, the higher mortality experienced by natives partnered with immigrants could be related to the disadvantages faced by the immigrant partner either via discrimination or a higher exposure to the pandemic (eg, having a frontline or precarious occupation). Although this type of exposure in mixed partnerships might not be at the same level as for immigrants partnered with another immigrant (as our results with a gradient in mortality suggests).

In this study, immigrants in different family constellations show higher levels of COVID-19 deaths than the majority of natives, after adjustment for a wide range of individual-level and contextual-level factors, including education, income, housing conditions and neighbourhood immigrant density. This set of adjustments also partly accounts for a number of socially patterned chronic health conditions and comorbidities, for example, insulin resistance, hypertension, smoking and obesity, which have been suggested as risk factors for severe cases of COVID-19.[17]

This study has a number of contributions and strengths, First, this is the only study to date to examine

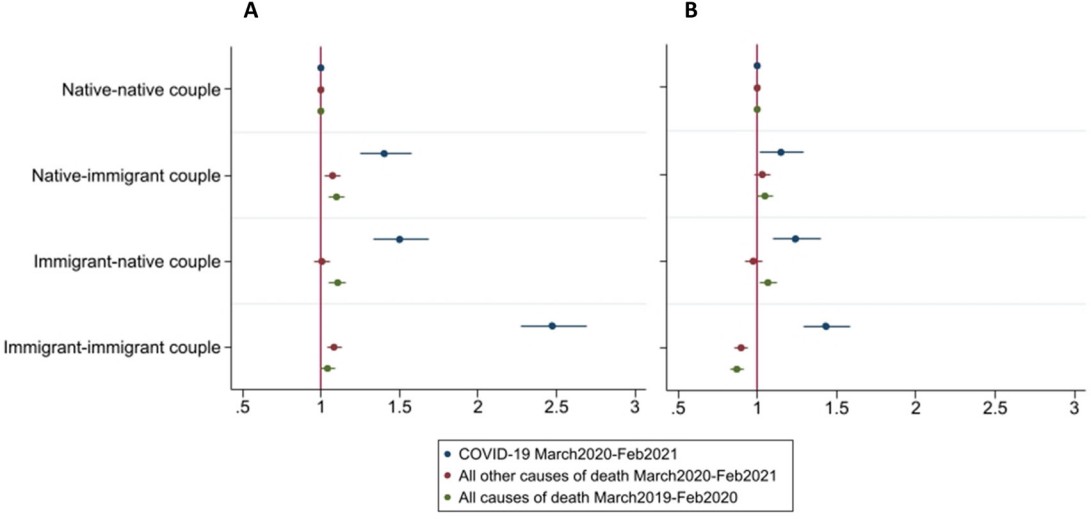

**Figure 2** HRs for the risk of dying from COVID-19, other causes of death during the pandemic, and all-cause mortality in the year prior to the pandemic by couple dyad. Model A is adjusted for age and sex only and model B includes full adjustment (reference group: native-native couples).

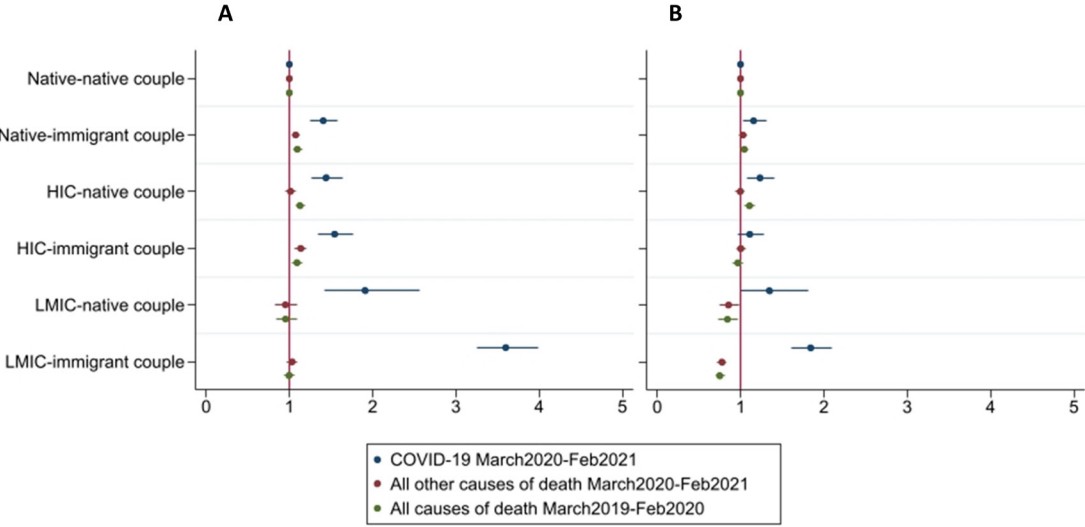

**Figure 3** HRs for the risk of dying from COVID-19, other causes of death during the pandemic and all-cause mortality in the year prior to the pandemic by couple dyad disaggregated by low-income and middle-income countries (LMIC) and high-income countries (HIC) immigrants. Model A is adjusted for age and sex only and model B includes full adjustment (reference group: native-native couples).

sociocultural integration as the mechanism behind the disproportionate burden that COVID-19 has placed on immigrant communities, by comparing the mortality of native-native, immigrant-native and immigrant-immigrant couples in Sweden. A major strength of our study is that we have complete coverage of the total population and all deaths in Sweden from the start of the pandemic until February 2021 for both COVID-19 and remaining causes of death. Thus, our analysis does not suffer from selection into our study population. We have similar high-quality data for the year prior to the pandemic, which allows for an unbiased comparison of mortality patterns between the 2 years. The comparison between couple types with respect to mortality from all other causes during the pandemic and all-cause mortality in the year prior strengthens our findings as it demonstrates that the excess mortality from COVID-19 is observed in couples that show no excess mortality from other causes before and during the pandemic.

Despite its strengths, this study has also some limitations that are worth mentioning. Although the Swedish population registers hold high quality and have many advantages, they capture *de jure* rather than *de facto* characteristics of individuals. With respect to our measure of partnership, 82% of the couples in our data are either married or have shared children while the remaining 18% co-reside unmarried without common children. However, non-marital cohabitation is very common in Sweden, while flat-sharing is not,[30] and particularly among individuals above their 30s. It is therefore fair to assume that a substantive share in this remaining group is indeed cohabiting in an amorous relationship, a group that is often overlooked in international studies of health and mortality because of the lack of data. A final limitation is that we do not include information on occupation. Specifically, it has been hypothesised that immigrants are

more likely to work in 'frontline' occupations; however, recent research in Sweden has shown that occupational exposure is not a risk factor for COVID-19 mortality.[31] Furthermore, it is worth noting that a majority of deaths are occurring among retired individuals with no attachment to the labour market.

In conclusion, our study shows that being partnered with a native does not close the gap in COVID-19 mortality with natives even after adjusting for a wide range of possible confounders on individual, couple and residential level. As such, these findings show that lack of awareness of the Swedish recommendations and language barriers are not major drivers for the excess COVID-19 mortality of immigrants. At the same time, the fact that Swedes partnered with immigrants also show excess mortality compared with Swedish couples, suggests that excess mortality among immigrants is explained by differential exposure to the virus.

**Acknowledgements** We thank Thomas Niedomysl of Region Halland, Petra Westin of the National Board of Health and Welfare and Simon Kurt of Statistics Sweden for their invaluable role in providing the data.

**Contributors** SA, MB and SJ conceived the study and were responsible for the planning. GA and MB provided the data. MB analysed the data. SA, SJ, SD, EM and MB designed the analysis with contributions from OO, MR, GA. SA, SJ, SD, EM, MB, OO and MR. GA contributed to the interpretation of the data. SA and SJ drafted the manuscript with substantive contributions from EM, SD, MB, OO, MR and GA. All authors approved the final version of the manuscript.

**Funding** The Swedish Research Council for Health, Working Life and Welfare (FORTE), grant numbers 2016-07115, 2016-07105, 2016-07128, 2019-00603, The Swedish Research Council (VR), grant number 2018-01825 and The Swedish Foundation for Humanities and Social Sciences (Riksbankens Jubileumsfond), grant M18-0214:1.

**Competing interests** None declared.

**Patient consent for publication** Not required.

**Ethics approval** This study was approved by the Central Ethical Review Board in 2020 (Dnr 2020-02199).

**Provenance and peer review** Not commissioned; externally peer reviewed.

**Data availability statement** This study is produced under the Swedish Statistics Act, where privacy concerns restrict the availability of register data for research. Aggregated data can be made available by the authors, conditional on ethical vetting. The authors access the individual-level data through Statistics Sweden's micro-online access system MONA.

**ORCID iDs**
Siddartha Aradhya http://orcid.org/0000-0003-3748-6270
Sol Pía Juárez http://orcid.org/0000-0001-9086-7588

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
