## [Reviewer comments · BMJ Open]

ARTICLE DETAILS

TITLE (PROVISIONAL)	Intermarriage and COVID-19 mortality among immigrants. A population-based cohort study from Sweden
AUTHORS	Aradhya, Siddartha; Branden, Maria; Drefahl, Sven; Obucina, Ognjen; Andersson, Gunnar; Rostila, Mikael; Mussino, Eleonora; Juárez , Sol Pia

VERSION 1 – REVIEW

REVIEWER	Indseth, Thor
REVIEW RETURNED	Norwegian Institute of Public Health, Health Services Research 18-Feb-2021

GENERAL COMMENTS	1. Summary of the research This manuscript argues that low levels of acculturation cannot explain the difference in COVID-19 mortality between immigrants and non-immigrants. The article bases its claim on the thesis that if COVID-19 mortality is related to lack of acculturation we should observe a difference in COVID-19 mortality between a) immigrants who are partners with immigrants and b) immigrants who are partners with natives and natives who are partners with immigrants. The article presents results that support this thesis. The data is register-based and include 836,390 Stockholm residents, with a total of 515 COVID-19 deaths, where 291 deaths are among native-native couples, 117 among immigrant-immigrant couples and 107 among native-immigrant couples. Differences in age composition within the groups are accounted for by using age as time scale. The article also adjusts for sex, and (in one model) for education and a set of other “neighborhood characteristics”. Two sensitivity analyses are conducted; one for country of origin as either high-income or low- and middle-income, and one in which natives with one or more foreign-born parent were excluded. The results show that rates for COVID-19 deaths in native-immigrant couples are about the same as rates for COVID-19 deaths among immigrant-immigrant couples. From this finding the authors conclude that their thesis is correct. From this the authors also conclude that “Public health strategies based on cultural differences might not only be inefficient but also reinforce stereotypes and health inequalities”. Uncertainties are mainly shown in the appendix. 2. Overall impressions
---

The main thesis is novel and the design is clever, but the sample (515) is too small to support the reported conclusions. Uncertainties seem to be highly downplayed and alternative explanations for the results are hardly explored or discussed. Uncertainties need to be stated more clearly, not only in the limitations-section but also in the abstract, results and conclusion sections. If the article is expanded to include a substantially larger data set and more rigorous methods are applied, then the results will be of high scientific interest and major societal relevance., also on an international level.

3. Specific areas for improvement (major importance)

- The major weakness of this paper is the small sample size, and the paper would benefit considerably if the sample size was increased by either extending the material geographically, temporally or (ideally) both. If this is not possible, the conclusion should be reformulated to “inconclusive”.

- Part of the conclusion is not supported by any data shown in this article: : “Public health strategies based on cultural differences might not only be inefficient but also reinforce stereotypes and health inequalities”. More specifically, no public health strategies are investigated or evaluated with any type of data. Neither is the question of how public health strategies affect stereotypes and/or health inequalities investigated by any type of data. This part of the conclusion is not substantiated and should therefore be removed.

- “Acculturation” is a key concept in this article, but it remains undefined. It is thus unclear what the authors actually refer to with this concept. “Acculturation” is a contested concept within several fields of social science and migration research. The authors should state how it is defined and what is included and excluded in the concept.

- It is unclear what the authors mean by “neighborhood characteristics”. What variables that are here adjusted for and not and where these data come from need to be accounted for in the methods section.

- The confidence intervals in the sensitivity analysis show that the uncertainties are too large for any conclusions to be made regarding the explanatory value of “country of origin being low-, middle- or high-income” and “second generation”. These explanations can therefore not be entirely disregarded.

- There are several highly relevant newly published studies on COVID-19 and immigrants from the Scandinavian context (NIPH and Statens Serum Institut) that are not included or discussed in this paper.

- The possible role of social milieu for transmission of COVID-19 should be discussed. E.g. the findings reported in this paper might be explained by native-immigrant couples having a higher degree of social interaction and frequenting the same social

	miliues as immigrant-immigrant couples, thus being exposed for some of the same vulnerabilities as do immigrant-immigrant couples.  - The article's thesis rests on the assumption that natives in the 'native-immigrant' couples hardly ever acculturates into the immigrant culture of their immigrant partner (thus being at increased risk). This assumption needs to be addressed and substantiated. 3. Specific areas for improvement (minor importance)  - The cutoff date (May 7th) seems arbitrary and should be accounted for. - It is worth investigating whether the use of "time scale" might skew the research results. What if one or more of the groups have a high number of 90+ year-olds (high risk group) but also a higher number of young? - Why do "language" and "understanding of the healthcare system" sort under "behavior" (page 5 line 52)? - The authors should show the size of N for the different sensitivity analyses (especially when second generation is excluded). - Why are "individuals who had not lived in Sweden in the two prior years (n=11,864)" excluded? Wouldn't this group be particularly interesting to study when investigating the effect of "acculturation"?
--	--

REVIEWER	Katsoulis, M University College London, Farr Institute of Health Informatics Research
REVIEW RETURNED	23-Feb-2021

GENERAL COMMENTS	This is an interesting paper showing that in Sweden, immigrants experience excess mortality relative to Swedes from COVID-19 across levels of acculturation. I have the following comments  1. The authors should add a table in the supplemental files where the descriptive characteristics of the individuals with and without missing values will be presented. This table will be very important to understand if the missing mechanism is missing completely at random (MCAR), missing at random (MAR), or missing not at random (MNAR) 2. Following up from point 1, there is an indication in the missingness mechanism is MAR, because the authors report in the methods, page 7 "Missing information on education is generally very low but 88% of those with missing education are immigrants"
--

So, the authors should perform multiple imputation or inverse probability of missingness weighting, to tackle the problem of missing values, because perhaps missing values on education might influence their findings. It also seems that the missing indicator for education is closely related to the outcome (14.64 deaths per 1000 years)

3. Tables and Figures should stand alone. The authors should clarify what they present in Appendix Figure 1.
e.g. Hazard Ratios adjusted for XXX between immigrant-native couple type and COVID-19 mortality
The same comment holds for all the tables and figures in the main text and the appendix. Titles like “Full regression table for Figure 3” are not informative.
Titles in figure 2 and 3 are not clear and need rephrasing

4. Minor comment: Flow chart should be shown appropriately, because both inclusions and exclusions are presented at the same flow; thus, it is misleading. Check STROBE guidelines on how to present the flowchart
<https://journals.plos.org/plosmedicine/article?id=10.1371/journal.pmed.0040297>

5. Minor comment: The authors should clearly present the 95Cis in Appendix tables 1 and 2

VERSION 1 – AUTHOR RESPONSE

REVIEWER 1#1: The major weakness of this paper is the small sample size, and the paper would benefit considerably if the sample size was increased by either extending the material geographically, temporally or (ideally) both. If this is not possible, the conclusion should be reformulated to “inconclusive”.

AUTHORS: Thank you for your comment. We now had the possibility of extending both the geographical context and the observation window. We now have expanded the analysis with data covering the total Swedish population and all COVID-19 deaths until February 23, 2021. The results have slightly changed after the expansion of the analysis, mainly as a result of narrower confidence intervals, but the conclusions largely remain.

REVIEWER 1#2: Part of the conclusion is not supported by any data shown in this article: “Public health strategies based on cultural differences might not only be inefficient but also reinforce stereotypes and health inequalities”. More specifically, no public health strategies are investigated or evaluated with any type of data. Neither is the question of how public health strategies affect stereotypes and/or health inequalities investigated by any type of data. This part of the conclusion is not substantiated and should therefore be removed.

AUTHORS: We agree that this part of the conclusion was not substantiated by our analyses and has now removed it from the conclusion.

REVIEWER 1#3: “Acculturation” is a key concept in this article, but it remains undefined. It is thus unclear what the authors actually refer to with this concept. “Acculturation” is a contested concept within several

fields of social science and migration research. The authors should state how it is defined and what is included and excluded in the concept.

AUTHORS: Thanks for this comment. We have carefully considered the use of acculturation and decided to be more precise regarding what aspects of this broad concept this study focuses on. In this study, we use intermarriage as an indicator of language abilities and institutional awareness, knowledge of the host country's institutions and social practices, and the ethnic composition of one's social circle, all of which have been considered as possible explanations for the excess COVID-19 among immigrants. In the revised version of the manuscript, we specify this as follows on page 5:

"Within the field of immigrant integration, intermarriage has been considered the ultimate stage of acculturation and is, both, a marker of and facilitator for socio-cultural integration. Marrying a native is strongly related to language abilities, knowledge of the host country's institutions and social practices, as well as the ethnic composition of one's social circle. As a result, intermarriage reflects the narrowing of socio-cultural distance between ethnic Swedes and immigrants, which renders the ideal measure to evaluate the role of understanding and awareness of recommendations as explanations for the excess COVID-19 mortality among immigrants."

For the sake of being clearer about the specific aspects of acculturation that we are capturing with intermarriage, we have decided to streamline the text to focus specifically on language and institutional awareness. This required a substantial rewrite of the text; however, the overall message has not changed.

"If understanding and awareness of recommendations (language in primis) explain the excess mortality, immigrants partnered with Swedes and in particular Swedes partnered with immigrants should have similar mortality to Swedes partnered with Swedes. To this end, the aim of this study is to examine the association between socio-cultural integration and COVID-19 mortality by examining native-immigrant couple dyads—a well-regarded measure that has been shown to be a marker and facilitator of integration".

This reasoning is now consistent over the text and our conclusion are strongly linked.

REVIEWER 1#4 It is unclear what the authors mean by "neighborhood characteristics". What variables that are here adjusted for and not and where these data come from need to be accounted for in the methods section.

AUTHORS: In the submitted version of the manuscript, we were unclear about what we meant by neighborhood characteristics. We have now further clarified that neighborhood characteristics refers to the immigrant density in one's neighborhood (DeSO). This is addressed in the manuscript on page 7:

"We include in our model also the share of immigrants in the local neighborhood, DeSO (a smaller subdivision in Swedish administrative statistics"

and on page 8:

"Two models were estimated: 1) a simple model with age as the time scale adjusted for sex and 2) the same model with further adjustments. In the latter analysis we adjust for education, household and immigrant density in the neighborhood, and county of residence fixed-effects that are confounders, as well as factors that are on the causal pathway that have been previously used to explain the excess

mortality of immigrants (i.e, number of individuals in the household below the age of 30, dwelling type, square meters per person in the dwelling, household income)".

REVIEWER 1#5 The confidence intervals in the sensitivity analysis show that the uncertainties are too large for any conclusions to be made regarding the explanatory value of "country of origin being low-, middle- or high-income" and "second generation". These explanations can therefore not be entirely disregarded.

AUTHORS: The new analyses include the total Swedish population and the period March 4, 2020 to February 23, 2021 which provides us with significantly more statistical power. Importantly, the results remain robust and support the initial explanations in the text. We think that these analyses are now even more convincing to show that 1) there is no effect heterogeneity among Swedes partnered with immigrants depending on whether their partner is born in HIC or LMIC and 2) the results are robust when second generation immigrants (individuals born in Sweden with foreign-born parents) are excluded from the analysis.

REVIEWER 1#6 There are several highly relevant newly published studies on COVID-19 and immigrants from the Scandinavian context (NIPH and Statens Serum Institut) that are not included or discussed in this paper.

AUTHORS: New references have now been included. See references 2 and 10. Specifically, we have included a recent study by the Indseth et al. 2020 and a systematic review that covers a number of recent studies, including those from the Statens Serum Institut.

REVIEWER 1#6 The possible role of social milieu for transmission of COVID-19 should be discussed. E.g. the findings reported in this paper might be explained by native-immigrant couples having a higher degree of social interaction and frequenting the same social milieus as immigrant-immigrant couples, thus being exposed for some of the same vulnerabilities as do immigrant-immigrant couples.

AUTHORS: Thank you for this comment and we fully agree that this was missing in the manuscript. We did not fully appreciate the possibility that natives partnered with immigrants experience different social contacts than natives partnered with natives. While editing the text to consider this, we were able to draw a stronger conclusion from our results that social milieus matter. Given that natives do not experience barriers with respect to language proficiency or institutional awareness and that biological vulnerability cannot be transmitted between partners, their elevated mortality is likely due to differences in exposure and the social vulnerabilities of their immigrant partner. This is now a more nuanced discussion in the text. Specifically we write the following on page 11 of the manuscript: "Beyond allowing us to disentangle the role of language barriers and lack of understanding of the healthcare system and recommendations in explaining the excess COVID-19 mortality among immigrants, our study provides suggestive evidence that differential exposure to the virus and not susceptibility is the culprit 28. Although it is true that Swedes partnered with a Swede show the lowest mortality, those partnered with an immigrant experience higher COVID-19 mortality. Given that Swedes do not experience language barriers or lack institutional awareness, and that biological susceptibility cannot be transmitted between partners, Swedes partnered with immigrants are likely to be at either higher exposure to virus or impacted by the social susceptibility of their partner. Swedes in mixed partnerships may be exposed to similar social environments and/or risk factors as immigrants that place them at a higher risk of exposure as compared to Swedes partnered with other Swedes. For example, they may have more transnational contacts or are impacted by some of the same risk factors as immigrants. Moreover, the higher mortality experienced by natives partnered with

immigrants could be related to the disadvantages faced by the immigrant partner either via discrimination or a higher exposure to the pandemic (e.g., having a frontline occupation or precarious occupations). Albeit this type of exposure in mixed partnerships might not be at the same level as immigrants partnered with immigrants (as our results suggests there is a gradient in mortality).”

REVIEWER 1#7 The article’s thesis rests on the assumption that natives in the ‘native-immigrant’ couples hardly ever acculturate into the immigrant culture of their immigrant partner (thus being at increased risk). This assumption needs to be addressed and substantiated.

AUTHORS:

Thank you for this comment because it made us think a lot more about what the results for mixed partnerships mean for our study. Now that we have more clearly defined what we are capturing by using intermarriage as a measure for language abilities, knowledge of the host country’s social practices, and the ethnic composition of one’s social circle, this is now addressed. As mentioned before (see response to comment 3) we tried to clarify our rationale for using intermarriage and, as a result, it is now less relevant to discuss acculturation as a two-way process. Specifically, natives in mixed partnerships are not experiencing excess mortality due to poor understanding of the health care system and public health recommendations, and/or fluency in the Swedish language. As such, we argue that natives partnered with immigrants may have different social networks and may be exposed to similar environments to immigrants. This finding further adds support to the hypothesis that these factors are not relevant for explaining the excess mortality of immigrants. Therefore, the higher mortality experienced by the native could be related to the disadvantages faced by the immigrant partner via either discrimination or a higher exposure to the pandemic (e.g., having a frontline occupation).

Specific areas for improvement (minor importance)

REVIEWER 1#8 The cutoff date (May 7th) seems arbitrary and should be accounted for.

AUTHORS: In the revised version of the manuscript, we use the most up-to-date data available to us from the Swedish Board of Health and Welfare. As a result, we are using the maximum amount of data available to us. The new analysis now covers the period March 4, 2020 to February 23, 2021.

REVIEWER 1#9- It is worth investigating whether the use of “time scale” might skew the research results. What if one or more of the groups have a high number of 90+ year-olds (high risk group) but also a higher number of young?

AUTHORS: Our model specification controls for the effect of age, which means that difference in the age structure between the groups do not affect our results. However, we could not capture the age effect only in the case of an interaction between the outcome and age for one or more of the groups but from the following histogram of the age distribution across the different groups in our analysis this does not seem the case.

REVIEWER 1#10- Why do “language” and “understanding of the healthcare system” sort under “behavior” (page 5 line 52)?

AUTHORS: Thank you for pointing this out. We have completely rewritten that section and this is now more accurately stated.

REVIEWER 1#11- The authors should show the size of N for the different sensitivity analyses (especially when second generation is excluded).

AUTHORS: These numbers are now included in the title of the figures.

REVIEWER 1#12- Why are “individuals who had not lived in Sweden in the two prior years (n=11,864)” excluded? Wouldn't this group be particularly interesting to study when investigating the effect of “acculturation”?

AUTHORS: This is just the nature of the data construction. Due to the immediacy of the research, we were required to merge COVID-19 mortality data to existing population registers updated until December 2019. Although this population would indeed be interesting to study from the perspective of acculturation, we are unable to do so due to data limitations. This has also been further clarified in the manuscript (see page 5).

Reviewer 2:

REVIEWER 2#1. The authors should add a table in the supplemental files where the descriptive characteristics of the individuals with and without missing values will be presented. This table will be very important to understand if the missing mechanism is missing completely at random (MCAR), missing at random (MAR), or missing not at random (MNAR)

AUTHORS: Thank you for your suggestion. We have attached the table here and included it in the appendix (table S6). The missing values do seem to be missing at random and have no impact on the distribution of other descriptive characteristics. We have also included a column where we exclude individuals with missing sqm per person in the dwelling to show that the distribution of the other covariates remains unchanged.

REVIEWER 2#2. Following up from point 1, there is an indication in the missingness mechanism is MAR, because the authors report in the methods, page 7 “Missing information on education is generally very low but 88% of those with missing education are immigrants”
So, the authors should perform multiple imputation or inverse probability of missingness weighting, to tackle the problem of missing values, because perhaps missing values on education might influence their findings. It also seems that the missing indicator for education is closely related to the outcome (14.64 deaths per 1000 years)

AUTHORS: We completely agree here that missing education is problematic for immigrants. We have followed the reviewer's advice and performed multiple imputation to test how the missing values for education impact our results. The results remain unchanged and the figure is now included in the appendix.

REVIEWER 2#3 Tables and Figures should stand alone. The authors should clarify what they present in Appendix Figure 1.

e.g. Hazard Ratios adjusted for XXX between immigrant-native couple type and COVID-19 mortality The same comment holds for all the tables and figures in the main text and the appendix. Titles like “Full regression table for Figure 3” are not informative. Titles in figure 2 and 3 are not clear and need rephrasing

AUTHORS: Thank you. We have now changed the figures and table titles.

REVIEWER 2#4. Minor comment: Flow chart should be shown appropriately, because both inclusions and exclusions are presented at the same flow; thus, it is misleading. Check STROBE guidelines on how to present the flowchart <https://journals.plos.org/plosmedicine/article?id=10.1371/journal.pmed.0040297>

AUTHORS: We have now updated the flow chart according to the STROBE guidelines.

REVIEWER 2#5. Minor comment: The authors should clearly present the 95Cis in Appendix tables 1 and 2

AUTHORS: We have now made this clearer.

VERSION 2 – REVIEW

REVIEWER	Indseth, Thor
REVIEW RETURNED	Norwegian Institute of Public Health, Health Services Research 04-Jun-2021

GENERAL COMMENTS	I appreciate the opportunity to review this paper. The authors have made substantial revisions, and earlier weaknesses and unclarities are now taken care of. Some minor, but important revisions are needed before publication. Comments: 1. The conclusion in the abstract is not supported by the results for at least two important reasons:a. The fact that there is a difference between N-N-couples and N-I/I-N-couples and between N-I/I-N-couples and I-I-couples in mortality actually supports that there is some unexplained factor. This hitherto unexplained factor might or might not be language and/or institutional awareness. Hence, the findings in the results section do not support an exclusion of language/institutional awareness as possible explanations; rather they support the conclusion that language/institutional awareness is not a major driver for excess mortality.b. It is possible that the relatively low difference between I-I-couples and I-N/I-couples can be, at least partly, a result of measures implemented to mitigate the negative effect of language barriers and low institutional awareness. I assume that also Sweden has translated and tailored information campaigns, and that interpreters have been used as a part of the Swedish Covid-19 response.2. A better base for the conclusion seems to be made in the point at page 11, second paragraph, first sentence, which is more in line with the findings.3. Page 10 Discussion, first paragraph, second sentence: This sentence is not supported by the findings in the results. The fact that having a Swedish born partner partially protects against Covid mortality is actually an argument for, not against, language and/or institutional awareness as a possible explanatory factor.4. Page 11, second paragraph: The authors should be open to the fact that also Swedish born individuals may lack institutional awareness.
---

5. Conclusion, page 13, second paragraph. Though more precise than the conclusion stated in the abstract, the main conclusion needs to be adjusted according to point 1 and 2 above.
--

VERSION 2 – AUTHOR RESPONSE

Reviewer: 1

Mr. Thor Indseth, Norwegian Institute of Public Health

Comments:

REVIEWER 1 #1: The conclusion in the abstract is not supported by the results for at least two important reasons:

a. The fact that there is a difference between N-N-couples and N-I/I-N-couples and between N-I/I-N-couples and I-I-couples in mortality actually supports that there is some unexplained factor. This hitherto unexplained factor might or might not be language and/or institutional awareness. Hence, the findings in the results section do not support an exclusion of language/institutional awareness as possible explanations; rather they support the conclusion that language/institutional awareness is not a major driver for excess mortality.

b. It is possible that the relatively low difference between I-I-couples and I-N/N-I-couples can be, at least partly, a result of measures implemented to mitigate the negative effect of language barriers and low institutional awareness. I assume that also Sweden has translated and tailored information campaigns, and that interpreters have been used as a part of the Swedish Covid-19 response.

AUTHORS #1: Thank you for your comment. We have now re-written the conclusion in the abstract to highlight that that language and institutional awareness are not major drivers for excess mortality. However, we would like to emphasize our results in our initial submission where we focused on the first stage of the pandemic in Stockholm, where translation of information was delayed. Specifically, we would have expected to find large mortality differences across groups, if language/institutional awareness were the explanations; however, we found no differences across I-I-couples and I-N/N-I-couples. In relation to point b, we have adjusted our discussion section in relation to the role of language and the public health measures implemented in this respect. See page #11 and 12.

REVIEWER 1#2. A better base for the conclusion seems to be made in the point at page 11, second paragraph, first sentence, which is more in line with the findings.

AUTHORS #2: Thank you. In the latest version of the abstract, we have included this idea. It reads now (page #11):

“Language barriers and/or poor institutional awareness are not major drivers for the excess mortality from COVID-19 among immigrants. Rather, our study provides suggestive evidence that excess mortality among immigrants is explained by differential exposure to the virus.”

REVIEWER 1#3. Page 10 Discussion, first paragraph, second sentence: This sentence is not supported by the findings in the results. The fact that having a Swedish born partner partially protects against Covid mortality is actually an argument for, not against, language and/or institutional awareness as a possible explanatory factor.

AUTHORS #3: Thanks. We have more clearly summarized these results. It now reads (page # 10):

“These findings challenge hypotheses that poor language proficiency and institutional awareness are major contributing factors explaining the excess mortality from COVID-19 among immigrants.”

REVIEWER 1#4. Page 11, second paragraph: The authors should be open to the fact that also Swedish born individuals may lack institutional awareness.

AUTHORS #4: Unfortunately we do not follow the reviewer’s rationale for this comment. All Swedish-born individuals have higher institutional awareness than comparable immigrants, even though variation within the Swedish population (for example by age and socioeconomic condition) is expected.

REVIEWER 1#5. Conclusion, page 13, second paragraph. Though more precise than the conclusion stated in the abstract, the main conclusion needs to be adjusted according to point 1 and 2 above.

AUTHORS 1#5: Thank you, we have re-written the conclusion according to points 1 and 2. It reads now (page #13):

“In conclusion, our study shows that being partnered with a native does not close the gap in COVID-19 mortality with natives even after adjusting for a wide range of possible confounders on both individual-couple- and residential level. As such, these findings show that lack of awareness of the Swedish recommendations and language barriers are not major drivers for the excess COVID-19 mortality of immigrants. At the same time, the fact that Swedes partnered with immigrants also show excess mortality compared to Swedish couples, suggests that excess mortality among immigrants is explained by differential exposure to the virus.”